# Twin-Screw Extrusion of Oat: Evolutions of Rheological Behavior, Thermal Properties and Structures of Extruded Oat in Different Extrusion Zones

**DOI:** 10.3390/foods11152206

**Published:** 2022-07-25

**Authors:** Chengyi Zhou, Min Wu, Dongyu Sun, Wenguang Wei, Haoze Yu, Tong Zhang

**Affiliations:** College of Engineering, China Agricultural University, No. 17 QinghuaEast Road, Haidian District, P.O. Box 50, Beijing 100083, China; sy20213071408@cau.edu.cn (C.Z.); sdymavis@163.com (D.S.); wwg5946@163.com (W.W.); B20203070556@cau.edu.cn (H.Y.); ziktlq@163.com (T.Z.)

**Keywords:** oats, extrusion zones, rheological behavior, starch pasting, structural variation

## Abstract

Further investigation of material properties during the extrusion process is essential to achieve precise control of the quality of the extrudate. Whole oat flour was used to produce low moisture puffed samples by a twin-screw extruder. X-ray diffraction (XRD), Scanning electron microscopy (SEM), infrared spectroscopy (FTIR), thermal analysis, and rheological experiments were used to deeply characterize changes in the structure and cross-linking of oats in different extrusion zones. Results indicated that the melting region was the main region that changed oat starch, including the major transformation of oat starch crystal morphology and the significant decrease of enthalpy representing the starch pasting peak in the differential scanning calorimeter (DSC) pattern (*p* < 0.05). Moreover, the unstable structure of the protein increased in the barrel and then decreased significantly (*p* < 0.05) after being extruded through the die head. The viscosity of oats increased in the cooking zone but decreased after the melting zone. A transformation occurred from elastic-dominant behavior to viscoelastic-dominant behavior for oats in the melting zone and after being extruded. This study provides further theoretical support for the research of the change of materials during extrusion and the development of oat-based food.

## 1. Introduction

Oat is a good source of minerals, proteins, polyphenols, and β-glucans [1]. The benefits of oats in reducing the risk of developing chronic conditions such as diabetes, obesity, and cardiovascular disease are well established [2]. However, oat food has been criticized for its storage, digestive absorption, and palatability, which require modification and optimization of its properties using the relevant processing technologies [3,4]. 

Extrusion-cooking is extensively used in the production of extruded snacks owing to its efficient and convenient processing [5]. In extrusion-cooking, the high temperature with high pressure treatment on food raw materials can maximize the digestibility of the components and reduce enzyme activity, besides the short duration time that can minimize the harmful effects on food such as browning and nutritional damage [6]. In comparison with other processing technologies such as gun puffing and microwave popping, few or no effluents are generated during the extrusion process due to their constant working condition, which results in a low-cost consumption [7]. Extrusion cooking allows great improvement in the physical and thermal properties as well as storage stability and digestibility of puffed snacks, accompanied by hydration, swelling, gelatinization, degradation and crystallinity variation of starch, protein denaturation, and formation of lipopolysaccharide complexes [8,9,10,11]. Extrusion cooking also improves safety because of the potential to reduce mycotoxin levels in cereals. For environmental concerns, extrusion has low production of waste water [12]. Moreover, due to the high degree of integration of extrusion technology, the production process has almost no pollutant emission, which is a green and low-carbon processing technology that is beneficial in the processing of grains and plant proteins. Currently, most regulations on the quality of the extrudate are mainly investigated from the aspect of the material’s formula and operating parameters (barrel temperature, screw shearing method, and moisture content) [13,14]. However, complex variations in the barrel of the extruder are ignored, which is a factor that directly affects the properties of the extrudate. This approach which ignores the extrusion process leads to poor understanding of the changes of material properties occurring within the extruder barrel and a lack of precision and generalizability in many research mechanisms [15]. For example, the actual temperature of materials reacting in the barrel may be higher than the setting barrel temperature due to friction between materials and screws as well as the tight packing of materials resulting from a high-speed feed rate [16,17]. Furthermore, variation in moisture content, screw speed, and temperature are related to fluctuations of pressure in the barrel, which has a direct impact on the expansion of cereals [18]. The difficulties in researching the changes of material inside the extruder, which is an enclosed environment with high temperature and pressure, result in the mutual reaction of components in the barrel being tough to obtain and the specific behaviors of materials that respond to the extrusion process being difficult to investigate experimentally.

“Dead stop” was proposed as a method which allows to investigate the reaction location and product transformation properties of different zones inside the extruder [19]. Chen et al. [15] investigated the chemical cross-linking and molecular aggregation of soybean proteins during extrusion with “dead stop” method and concluded that the hydrophobic interactions, hydrogen bonds, disulfide bonds, and their interactions during extrusion together varied the structure of the extrudate according to the analysis of protein solubility. CAI et al. [19] collected starch in an extruder barrel, analyzed the variations in starch during extrusion, and developed a rate equation to predict the starch gelatinization behavior process through a comprehensive analysis of the residence time in each section and degree of gelatinization. Other researchers concluded that the interface of the die and cooling zone is the position that affects the formation of meat-like fibrous by observing the process of high moisture extrusion using low-field nuclear magnetic resonance, SDS-PAGE, and linear rheological behavior [20]. Based on the setup conditions, each zone inside the extruder has different effects on the material. Materials in the feeding zone, namely the native oat, fed at a consistent rate by the feeder machine. In the mixing section, with the moisture input, water and powdery materials intermingle with each other and form a dispersion phase, followed by the friction between the materials being reduced and their aggregation being promoted [21]. Materials in the cooking area are subjected to higher temperatures and stronger screw shearing, resulting in a less stable structure with cross-linking reaction and shearing fragmentation [5]. According to a previous study, the oxidation of lipid is favored by increasing temperature above 130 °C, whereas a temperature range of 150–180 °C is more effective for the binding of lipids and other components [22]. The barrel is completely filled with molten fluid in the melting zone, where the molecular chain of the material is completely open, forming a disordered structure [19]. Thus, the melting zone is the section where the polymer accumulates, and the reaction is concentrated. Oat flour is conveyed through the round die of the extruder via screws, and the superheated pressure inside the oats is transformed into atmospheric pressure, which suddenly releases the accumulated steam, generating an expanding granular matrix with a porous structure and a series of transformations in their composition. However, most research studies have been conducted on single component substances, while oats as a whole grain are subject to complex interactions among its components during extrusion, and “dead stop” was used in this study to investigate the variation of oat in extruder in order to supply support for the development of new oat puffed foods. 

Rheology refers to the study of material flow and is used to explain the response of materials to imposed forces and deformations. In the case of food processing, the study of the rheological properties of food systems significantly contributes to optimizing the production process as well as the stability and taste of the final food [23]. Variation in the structure of oats at different stages of extrusion is a direct reason for the change in viscosity, and the viscosity change of the melt in the extruder is associated with the growth and incorporation of bubbles in the polymer, which have an impact on the puffing effect of the extrudate [24]. Chen et al. [20] investigated steady-state shear and frequency sweep experiments on protein-starch mixtures in different sections of a barrel and revealed a reduction in the impact of extrusion and shearing on viscosity. In rheological experiments, in comparison with linear rheological behavior, large amplitude shear experiments can better reflect the microstructural differences in complex fluids and are more appropriate for the variation state of oral processing [25]. A notable case is that glutenin and gliadin show different large amplitude oscillation shear (LAOS) behaviors owing to their binding structure and network behavior [26]. Furthermore, the content of β-glucan, crude fiber, and total carbohydrates has been demonstrated to have an impact on LAOS behavior, in accordance with the study by Carriere et al. [27]. Hence, it is feasible to investigate the relevant transformations in the process of oat extrusion by analyzing the results of rheological experiments.

This study aims to investigate the variation regulations of complex grains in the extruder through analyzing the structural changes and rheological properties of native oat, extrudate, and oat in three zones of a barrel through “dead stop” experiments, which is important for the precise regulation of product quality. Rheological experiments were performed to investigate the viscoelastic changes in the oat system. X-ray diffraction (XRD) was used to represent the crystal structure of the mixed system, DSC to reflect the variations in the thermodynamic properties, and Fourier transform infrared spectroscopy (FTIR) and SEM to observe the transformation of oats in the microstructure.

## 2. Materials and Methods

### 2.1. Materials

Oat flour obtained from Hogu Food Co., Ltd., China was used as the native material for extrusion experiments. The chemical compositions of the native materials are listed in Table 1.

### 2.2. Extrusion and Sample Collection

A co-rotating twin-screw extruder (TwinLab-F 20/40, Brabender, Germany) equipped with a round strand die head at the exit of the extruder barrel was used for the sample pretreatment. The five temperature-controlled sections were set to 50 °C, 70 °C, 100 °C, 120 °C, 140 °C, and 160 °C, respectively, so as to follow the transformation of oats over a broad temperature interval. The oat flour was fed into the extruder at a constant speed of 200 rpm (3.36 kg/h), the water speed was maintained at 18.9, 27.3, and 33.3 rpm (0.3, 0.43 and 0.53 kg/h) to ensure that the final moisture content was maintained at 15%, 18%, and 20%, respectively, and the screw speed was kept constant at 140 rpm. When homogeneous products were extruded and the pressure inside the barrel reached a steady state, the extruder was turned off and the barrel was opened quickly. As shown in Figure 1, samples from the mixing, cooking, and melting zones were collected followed with being lyophilized and stored at room temperature for further analysis.

### 2.3. X-ray Diffraction (XRD)

Changes in the crystal type and crystallinity of the starch samples were detected by X-ray diffraction (XD-2, Beijing Purkinje General Instrument Co., Ltd., Beijing, China), with a detector (D/tex ultra, 36 kV and 20 mA) and a Cu Kα radiation wavelength (λ = 1.5406 Å). The samples were evaluated from 5° to 40° (2θ scale) using a step size of 0.02°. The diffractograms were analyzed and the relative crystallinity of the sample was obtained from the ratio of the area of the crystalline region to the total area of the X-ray diffraction pattern using MDI Jade 6 software (MDI, Livermore, CA, USA) [28].
(1)$$X_{C}(\%)=\frac{F_{K}}{F_{K}+F_{A}}\times 100$$
where *X_C_* is the crystallinity, *F_K_* is the area of the crystalline region, and *F_A_* is the amorphous area.

In XRD patterns, an A-shaped crystal structure for oat starch is expected, with peaks in 5°, 17°, 18°, and 23°. During the extrusion process, a transformation from A-shaped to V-shaped crystal structure is expected for starch pasting [29].

### 2.4. Thermal Properties

The thermal properties of the oats at different extrusion stages were measured using differential scanning calorimeter (DSC-Q20, TA Instruments, New Castle, DE, USA). Each sample of 3 ± 0.3 mg was weighed in an aluminum pan, followed by the addition of water to obtain a sample: water ratio of 3:7. An empty pan was used as the reference. The samples were equilibrated for 12 h at room temperature prior to measurements. The first scanning temperature range was from 0 to 130 °C and the heating rate was set constant at 10 °C/min, then cooled back to 0 °C at 20 °C/min, and then reheated from 0 to 130 °C at 10 °C/min. Thermal parameters were defined as peak temperature (*T_p_*) and enthalpy change (*ΔH*).

In DSC patterns, a peak near 65 °C representing the heat absorption of starch pasting, a peak near 95 °C for changes of lipids, and a peak near 110 °C reflecting the denaturation of protein is intended [30].

### 2.5. Fourier Transform Infrared Spectroscopy (FTIR)

The functional groups of the oats at different stages were measured using Fourier transform infrared spectroscopy (Spectrum 100, PerkinElmer, Waltham, MA, USA). Approximately 2 mg of each sample was ground with 200 mg potassium bromide (KBr) and pressed into thin slices. The experiment was performed in transmissive mode with 32 scans and the data were averaged for each run with a resolution of 4 cm^−1^ over a range of 4000–400 cm^−1^.

Peak Fit v4.12 (SeaSolve, San Jose, CA, USA) was used to fit the obtained curves in Amide I (1600~1700 cm^−1^) by deconvolution with second-order functions to obtain the changes in the secondary structure of the oat protein after extrusion.

In FTIR patterns, functional groups around 3400 cm^−1^ represent O-H stretching vibrations for intra- and inter-molecular hydrogen bonding, functional groups in the range of 1600–1700 cm^−1^ (Amide I absorption) represent C=O stretching for protein, and functional groups in the range 900~1200 cm^−1^ represent stretching vibration of C=C and C-O for carbohydrates are represented.

### 2.6. Rheological Behavior

The sample (10 g) was accurately weighed, evenly mixed with 100 mL of deionized water, and equilibrated at room temperature for the measurement. All rheological experiments were performed using a TA ARES—G2 rheometer (TA Instruments, New Castle, DE, USA) at 25 °C after the gap calibration. A 50 mm APS parallel plate (Stainless steel HB) was used, and the measuring gap was set to 1 mm. In addition, a circle of silicone oil was dropped around the parallel plate before testing to prevent sample moisture evaporation. Moreover, Power Law and Cross models were used to fit rheological curves.

#### 2.6.1. Large Amplitude Oscillation Shear Measurement (LAOS)

A fixed frequency of 1HZ and a strain range of 0.1–1000% were set. Storage modulus (*G′*) and loss modulus (*G″*) of oat paste as a function of stress were recorded and 15 data points were taken in each logarithmic interval at 25 °C.

#### 2.6.2. Small Amplitude Oscillation Shear (SAOS) Measurement

The frequency sweep test was performed at an angular frequency ranging from 1 rad/s to 100 rad/s within the linear viscoelastic limit (0.3% strain). Consequently, the curves of the storage modulus (*G′*), loss modulus (*G″*), and loss factor (*tanδ*) with frequency were obtained. The loss factor *tanδ* represents the ratio of the viscous part to the elastic part, which can be calculated using Equation (1) [31].
(2)$$tan\delta =\frac{G^{\prime \prime }}{G^{\prime }}$$

The frequency dependence can be analyzed by a Power Law model using Equations (2) and (3) [31].
(3)$$G^{\prime }=K^{\prime }\times \omega^{n^{\prime }}$$
(4)$$G^{\prime \prime }=K^{\prime \prime }\times \omega^{n^{\prime \prime }}$$
where *n′* and *n″* represent the frequency modulus exponents and *K′* and *K″* represent the model parameters (Pa/sⁿ).

#### 2.6.3. Flow Behavior Measurement

The steady-state shear test was conducted at 25 °C with a shear rate range of 0.1–100 s^−1^. Power Law and Cross models were used to fit the shear rate *γ* (1/s) and viscosity *η* (Pa∙s) as Equations (4) and (5).
(5)$$\eta =K\times \gamma^{n-1}$$
(6)$$\frac{\eta -\eta_{\infty }}{\eta_{0}-\mu_{\infty }}=\frac{1}{1+{\left(\lambda \gamma \right)}^{m}}$$
where *K* represents the consistency coefficient, *n* represents the flow behavior index, *η*_0_ and *η_∞_* indicate the viscosity values when *γ*→0 and *γ*→∞, *λ* is the consistency of the Cross model, and *m* is the rate index.

### 2.7. Scanning Electron Microscopy (SEM)

The morphology of the samples was observed using SEM (SU3500, Hitachi Int., Tokyo, Japan) at an accelerating voltage of 15.00 kV. The samples were spread on double-sided conductive adhesive tape attached to a specimen holder and coated with gold (10 nm) for 60 s. Images were captured at 1000× magnification.

### 2.8. Statistical Analysis

Origin (Version 2021, Origin Lab Corporation, Northampton, MA, USA) was used for data analysis, plotting, and statistical evaluation. Raw data analysis was performed using MITlaos software (MATLAB R2016a, Natick, MA, USA). All data were tested in triplicate, represented as the mean ± standard deviation (SD) and statistically analyzed using one-way ANOVA in SPSS (version 24.0, SPSS Inc., Chicago, IL, USA) with a 95% confidence level. *p* < 0.05 was considered as significant difference.

## 3. Results

### 3.1. X-ray Diffraction (XRD)

According to composition measurements, native oats contain more than 70% carbohydrates, mainly starch, which has been shown to have a crystalline structure [32]. The XRD patterns of the oats in the five extrusion stages with 20% water content are shown in Figure 2. The positions of the diffraction peaks of the samples at the different treatment stages are listed in Table 2. Clear diffraction peaks at 13.4°, 16.3°, 18.3°, and 21.3° were observed for the untreated oat flour, indicating that the crystalline form of oat flour is A-type [33]. Oats in the mixing zone have a crystal structure comparable to that of the native material, which demonstrates that no significant changes in oats occur in the mixing zone. The reason for concern is that the shear strength of the screw in the mixing zone is not intense, only as a mix and transport the role of the material, and the barrel temperature is not enough to provide conditions for the reaction of materials. The barrel temperature in the cooking zone had increased to 100 °C, which exceeded the temperature of starch pasting, but the XRD patterns did not change largely. It was found that the oats in the cooking zone still existed in powder form by opening the extruder. The color of oats in the cooking zone was obviously darker compared with oats in the mixing zone, but the moisture in the barrel did not fully react with the materials. It can be concluded that the increase in temperature did not cause the transformation of the crystal structure of oat starch.

Oats in the melting zone have formed a fluid form of polymer with water at high temperatures. The number of diffraction peaks decreases in the melting zone; a small peak at 17° and a distinct peak at 19° were observed before entering the round die head. As the sample was ejected and extruded, the diffraction peak disappeared and the XRD patterns became smooth. Finally, a V-type crystalline structure was observed after extrusion [29]. The original crystal structure of the starch was destroyed by high temperature and pressure of extrusion. As a result, the particle size of the starch crystals decreased, and the periodicity of the lattice was disrupted. After the distortion of the lattice, the X-ray diffraction peaks are shifted, which is reflected in the graph in which the diffraction peaks gradually become wider, and finally some of the diffraction peaks disappear. Additionally, the typical crystal type of starch-lipid complexes is a V-type structure, verifying the bonding of starch and lipids during extrusion [34]. As shown in Table 3, the relative crystallinity of the samples showed a gradually decreasing trend with the advancement of the extrusion section, which proved that the extrusion process was related to the dextrinization and degradation of starch, resulting in the destruction of its crystal structure [35]. Moreover, Hoover et al. [36] revealed that starch and lipid complexes contributed to an impact on the V-shaped crystal structure.

There were no significant differences in the moisture content from 15% to 20% among the treatments. This could be explained by the fact that an overly high temperature might have caused a high velocity water loss, which weakened the effect of moisture variation on the oats.

### 3.2. Fourier Transform Infrared Spectroscopy (FTIR)

FTIR is a widely used method for evaluating interactions between proteins and carbohydrates. Figure 3 shows the variation of different oats in the infrared spectrum. The FTIR spectra indicate the absence of new chemical bonds in the samples during extrusion. Oat flour is a mixture of starch, protein, cellulose, and other compounds, and its FTIR spectrum exhibits the characteristics of multiple substances. Appendix A lists the chemical bonds of oats during extrusion. The correlated peaks of cellulose are located in the ranges of 3200~3600 cm^−1^, 1200~1266 cm^−1^, 1020~1030 cm^−1^, and 860 cm^−1^ [37]. The absorption peaks at 3340 cm^−1^ and 1023 cm^−1^ showed fluctuations, indicating the enhanced stretching effect of the O-H and C-O structures. Peaks at 1600~1700 cm^−1^, 1480~1575 cm^−1^, and 1373 cm^−1^ were assigned to proteins [38]. The absorption peak at 1657 cm^−1^ is related to the variation in the stretching of C=O in Amide I during the extrusion process. The peaks for carbohydrates were primarily located in the range 900~1200 cm^−1^. The changes around 1155 cm^−1^ are connected with the stretching vibrations of the C=C and C-O groups of the starch chains [39]. Moreover, the peaks characterizing the irregular stretching of the C-H bond of the methylene group (2927 cm^−1^) varied during the extrusion process.

As oats are characterized by their high protein content compared to other cereals (12.6%), Table 3 demonstrates the changes in the secondary structure of oat proteins during extrusion. Notably, α-helices and random coils showed an increasing trend in the melting zone, while β-sheets or β-turns showed an increasing tendency for extrudates. These observations suggest that α-helices are not stable in the environment of high temperature and strong shearing and the transformation from an unstable to a stable structure is consistent with the conclusions of Lisiecka et al. [40].

### 3.3. DSC

The thermal properties of the extruded oat samples were characterized using DSC measurements. As shown in Figure 4, three peaks located near 65 °C, 95 °C, and 110 °C can be captured in the DSC scan of untreated oats. According to the research of Paton et al. [41], the first peak at 62~65 °C represents the loss of starch crystallinity with its pasting in excess water. The peak at 95 °C indicated the transformation and melting of the complex formed by amylose and lipids, and the formation of the third peak at 110 °C was due to the denaturation of oat protein. A secondary scan of untreated oats revealed that only the peak at 95 °C was left, and the enthalpy of the secondary scan showed an increasing trend, indicating that heating intensified the reaction of amylose-lipid complex formation; this result is consistent with the study of Moisio et al. [30].

According to Table 4, as the extrusion process progresses, the enthalpy of the peak indicating starch pasting (65 °C) becomes significantly smaller after the cooking section, which means that the heat pasting of starch occurs mainly in the mixing and cooking zones in the extruder. The enthalpy of peak 2 and peak 3 reflecting the amylose-lipid mixture (95 °C) and the denaturation of oat protein did not show significant changes, but the position of the terminal screws (section III and IV) showed a lower enthalpy compared with other sections, which confirms the conclusion that extrusion and heating have a negative effect on the formation of the two compounds and lead to the degradation of lipids [42]. The denaturation of oat globulin is induced by the extrusion and puffing action of the extruder. With the heating action of the extruder and rotational shearing of the screws, the amino acid molecular chains of the oat protein were opened and finally reorganized in the die opening area. During this process, oat globulin is denatured and its secondary structure changes. Simultaneously, the conditions for protein formation in fibrous structures become difficult under the influence of starch [43]. In the second scan of all samples, only the peak at 95 °C remained and showed an increase in enthalpy, indicating an increase in the number of amylose-lipid compounds. These results can be attributed to the first heating step, which promotes complexation between lipids and amylose [44].

### 3.4. Rheological Behavior

#### 3.4.1. Flow Behavior

As shown in Figure 5a, the viscosity decreased with increasing shear rate, demonstrating that the oat samples exhibited shear thinning behavior as pseudoplastic fluids. Because of the breakdown of the lattice structure of a dispersion system composed of oat flour and water with increasing shear speed, this leads to a decrease in the viscosity [45]. Kristiawan et al. [46] observed a similar phenomenon for pea flour. At the same shear rate, the viscosity of the oats showed a decreasing trend from the feeding section (raw) to the mixing section (section I). After the second heating section, the viscosity increased slightly and then declined continuously after the oats passed the melting section (section III) and die head (section IV) under high pressure extrusion. This was because the mixing of water and oats increased the water content of the material, which diluted the concentration of the oat paste and reduced the viscosity of the material [47]. Subsequently, the starch of the oats and water under the action of high temperature underwent a gelatinization reaction after the second stage of heating, resulting in an increase in viscosity [48]. The decrease in viscosity in the third and fourth sections may be due to the degradation of starch at excessive temperatures [49]. The slope of the shear curve shows a decreasing trend, indicating a reduction in the degree of shear thinning.

For a more intuitive reflection of the sample trend, the shear curve was fitted using the Power Law and Cross models in Table 5 and a better fit of the Cross model (R^2^ > 0.99) was obtained. From the perspective of the Power Law model, fluctuations in K reflect changes in viscosity, whereas changes in n reflect changes in the degree of shear thinning of the sample [50]. From the perspective of the Cross model, the zero-rate viscosity η_0_ and consistency λ, which reflect the microstructure of the sample, were in accordance with the variation pattern of viscosity [51]. Furthermore, the rate index m showed a progressive increase in viscosity with extrusion over time, indicating that the viscosity is in greater dependence on the shear rate or more influenced by the shearing action. Mutual proximity of infinite-rate viscosity occurred for the η_∞_ values for each sample. This may be attributed to the yield stress provided by the hydrogen bonding between the starch granules in the suspension. At low shear rates, the formation of particle lattices and deformation cannot be quickly balanced. In contrast, the particles can be quickly aligned in the shear direction at high shear rates and simultaneously form a mesh structure that is dynamically balanced with shear deformation [52].

#### 3.4.2. SAOS Behavior

The internal structures of the oats in the different extrusion sections were revealed using a frequency sweep test. Variations in the storage modulus (*G′*), loss modulus (*G″*), and loss tangent *tanδ* as a function of frequency are shown in Figure 5b. *G′* reflects the elastic-like (solid) behavior and *G″* reflects the viscous-like (fluid) behavior of the samples [53]. All samples presented a high frequency dependence with *G′* and *G″* being positively correlated with angular frequency ω, which illustrates that the oat network structure is primarily composed of physical cross-linking of non-covalent bonds [54]. The rates of variation of *G′* and *G″* remained approximately constant in the Log coordinate system, suggesting that extrusion did not affect the degree of frequency dependency of different oats [55].

All samples exhibit the characteristic *G′* > *G″*, which is confirmed by the fact that *tanδ* < 1. Based on this phenomenon, it can be deduced that the elastic part of the sample is larger than the viscous part and all samples demonstrate gel-like behavior [56]. When the angular frequency remained constant, the *G′* and *G″* values of the samples at different squeezing stages showed a downward trend, indicating a decreasing viscosity and elasticity of the oat paste. Mechanical damage due to high temperature and pressure leads to the breakage of intramolecular hydrogen bonds and covalent bonds, which degrade starch macromolecules to generate small molecule oligosaccharides, resulting in a lower viscosity of the samples [57]. On the other hand, this declining trend can be assumed that the degradation of oat molecules by extrusion and shearing is irreversible.

Owing to the high temperature and pressure of extrusion, the loss angle *tanδ* of the sample progressively increased (Figure 5c), and those of the samples in the die head area and extrudates were evidently greater than those in the other stages. The loss angle of the oats reached the maximum value in the die head area, indicating a decline in the gel-like behavior of the sample, and the samples in the die head area approximated the most fluid. This phenomenon is attributed to the accumulation and longest residence time of the oats in the barrel. At this stage, oats are subjected to a series of reactions, which are pasted and degraded, along with a decrease in β-glucose content coupled with cellulose degradation, denaturation of globular proteins, and cross-linking reactions of lipids and straight-chain starch. As a result, the water-soluble material of oats was enlarged, and the viscoelastic ratio was altered to obtain a substance closer to the characteristics of fluid-like behavior.

According to the fitted Power Law function in Table 6, *K′* and *K″* follow a distinct tendency to decrease as extrusion proceeds, with the maximum values of *n′* and *n″* occurring in the die mouth region. For section III and IV, *n* is larger than in other regions, indicating that the effects of extrusion and expansion lead to an increased sensitivity of the frequency dependence.

#### 3.4.3. LAOS Behavior

Information on the structural deformation of complex systems under large distortions can be provided by nonlinear rheological properties, which have been used to analyze the relationship between the rheological behavior and microstructure of complex fluids [58]. The experimental results of the LAOS behavior for the five extrusion degrees of oat flour are illustrated in Figure 5d. Within the linear viscoelastic region, both *G′* and *G″* of all samples were fairly constant, with no variation with the enhancement of the oscillation strain. With the entry of the oat paste into the nonlinear viscoelastic region, both *G′* and *G″* exhibited a decline with an increase in strain. An explanation for this behavior can be derived from the theory of strain thinning (I) [59]. Under conditions of high strain, the rate parameters of the formation and decomposition of the oat network structure are negative, whereas the rate parameter of decomposition is positive [60]. The difference between them led to a rapid disruption in the network structure of the sample. Simultaneously, the molecular chains of oat paste were disentangled and gradually formed an arrangement in line with the flow direction, which resulted in the reduction of intermolecular entanglement with easier flow and a lower modulus [59]. The loss modulus of the oat pastes without any extrusion exhibited an enhancement with a small peak when they entered the nonlinear viscoelastic zone. This phenomenon was interpreted as weak strain overshoot (III). This phenomenon could be attributed to the sensitive interactions between the suspensions in the oat paste, leading to intermolecular entanglement and a partial increase in modulus [61]. This phenomenon was eliminated by the mixing effect of the screws.

In Figure 5d, a decrease in *G′* and *G″* of oat paste in five different sections of the extruder barrel reveals that mechanical shearing and high temperature cooking reduced the intensity of the network structure of oats [62]. The molecular chains were opened in the barrel and restructured in the die head. Oats in the melting zone, which was the end of the barrel, obtained the loosest network structure and the smallest *G′* and *G″* [63].

#### 3.4.4. Lissajous Curves Analysis

The results from the LAOS experiment were decomposed into elastic and viscous components in the Lissajous curves, which were used to analyze variations in the elastic and viscous properties within a single system [64]. According to the Lissajous curves (Figure 6a,b), the black line represents the total stress, and the red line indicates the elastic (Figure 6a) or viscous component (Figure 6b). Consistent with previous studies, the total stress line appeared oval in shape when the strain was less than 1%, exhibiting characteristics related to the linear viscoelastic region [65]. Deformation of the total stress line occurred as the strain was enhanced, as evidenced by the elastic stress curve deviating from the total stress. The area contained in the black curve gradually grew to become virtually rectangular, whereas the viscous stress curve approached the total stress. This phenomenon marks the transition from elastic-dominant behavior to viscous-dominant behavior with the increase in strain, and eventually manifests itself as plastic behavior [66]. Schreuders et al. [66] interpreted this as rupture of the structure in the LAOS test and viscous dissipation.

By comparing the changes of the Lissajous curves in different extrusion stages, it can be found that the total stress curve of oat in the extruder is larger than native oats, which indicates the increase of moisture and barrel temperature has a positive effect on the increase of viscoelasticity. The viscoelasticity of oat in the cooking zone is consistent with that of oats in the mixing zone. However, the elastic stress curve and the viscous curve of oats in the melting zone are transformed into a standard oval, indicating that oats transformed from an elastic-dominance behavior to a viscoelastic behavior in the melting zone [65]. In addition, a reduction is observed in the areas of oats in the melting zone and extrudate (sections III and IV), illustrating the energy curtailment in the oscillation process and decrease in viscoelasticity [67]. This phenomenon can be attributed to the disruption of starch and breakage of molecular bonds that maintain a stable structure during thermal extrusion [68].

### 3.5. Scanning Electron Microscopy

The microstructure of oat flour in the five extrusion regions was investigated, as shown in Figure 7. Native oats appear in the form of grains of different sizes, which are irregular in shape, mostly round, oval, and polygonal, with a smooth surface. The oat grains are adhered to each other, which is due to the possible action of protein and water in oats. The granules tended to aggregate in the mixing zone, which is related to the addition of water. The adhesion of oats in the mixing zone was more pronounced compared to the raw material, but the morphology was not altered, which verified the results of the similar properties of these two regions in XRD and DSC. In the cooking zone, the original round shape transformed into large aggregates, which were considered to be starch-lipid complexes formed by amylose and lipids at high temperatures. Meanwhile, the smooth surface of the oat grain becomes rough and broken particles present on the surface [30]. The samples in the cooking zone have already shown some resemblance to the final shape of the extrudate, but the aggregated state has not been fully formed; it may be due to enhanced mixing from the high temperature and pressure environment, which catalyzed changes in the protein [69]. In the melting zone, oats have completely changed the spherical particle state. Oat flour appeared as a sponge with a rough surface in the effect of protein cross-linking. As the oats flew out of the die head, the internal water evaporated rapidly owing to the large pressure difference, eventually resulting in a porous and loose structure.

## 4. Conclusions

This study focuses on the microstructural characteristics and rheological behavior of oats in different extrusion zones. Raw oats and oats in the mixing zone share similar thermal properties, functional groups, and crystal structures. Moisture input and screw shearing reduce the viscoelasticity of oats to a certain extent while promoting the aggregation of starch granules. In the cooking zone, a partial transformation of crystal structure occurs from A-type to V-type and the stable β-turns in oat globulin transform into unstable α-helices; the viscosity showed an upward trend at elevated temperature. Moreover, the solutions varied from a weak strain overshoot (III) to strain thinning (I) in this section along with viscoelasticity reduction and energy curtailment. The melting zone is the area where the compound fills the barrel and major reactions occur. Only the peak near 18° remains in the XRD profile of oat starch. Starch pasting and degradation of lipids occur in this region, with the highest percentage of proteins with unstable structures. Meanwhile, oat in the melting zone suffers the highest pressure in the barrel; rheological experiments show that the oat samples possess the lowest viscoelasticity and the loosest network structure here. When the extrudate is ejected, the internal water evaporates rapidly under the action of a huge pressure difference, forming a porous structure, which is related to a decline of relative crystallinity. The unstable α-helices in oat globulin transform into β-sheets and β-turns. This research provides further theoretical support for the development and quality control of oat-based healthy food.

## Figures and Tables

**Figure 1 foods-11-02206-f001:**
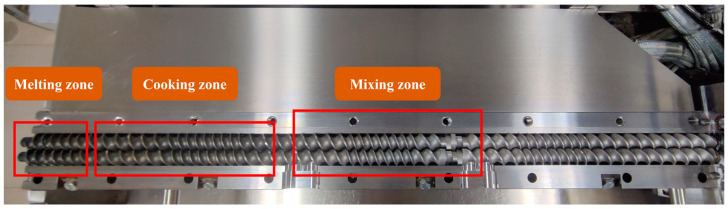
Position of mixing zone, cooking zone, and melting zone in barrel of extruder.

**Figure 2 foods-11-02206-f002:**
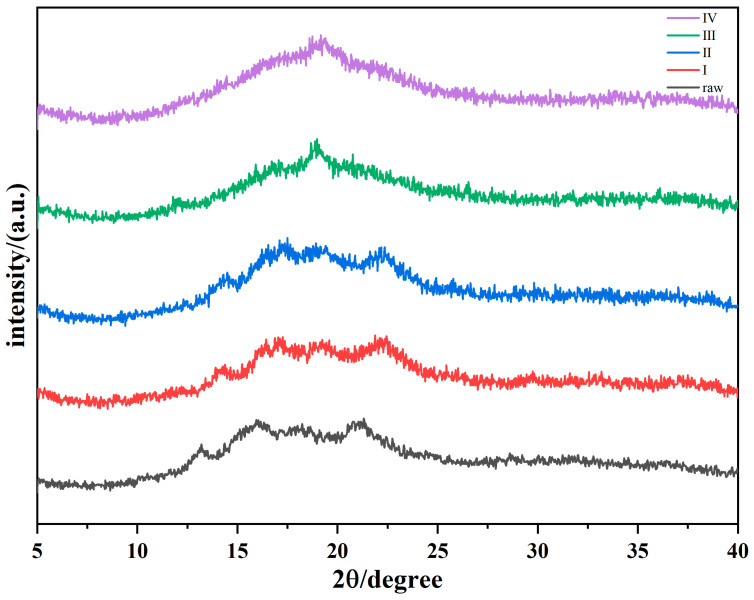
XRD pattern of oats from different zones at 20% moisture. Raw: native oat; I: oat in mixing zone; II: oat in cooking zone; III: oat in melting zone; Ⅳ: extrudate.

**Figure 3 foods-11-02206-f003:**
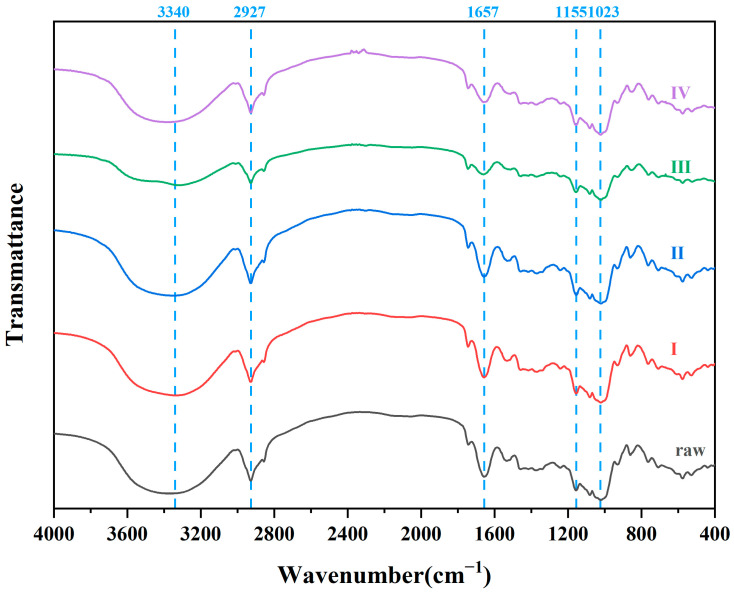
FTIR patten of oats from different zones at 20% moisture. Raw: native oat; I: oat in mixing zone; II: oat in cooking zone; III: oat in melting zone; Ⅳ: extrudate.

**Figure 4 foods-11-02206-f004:**
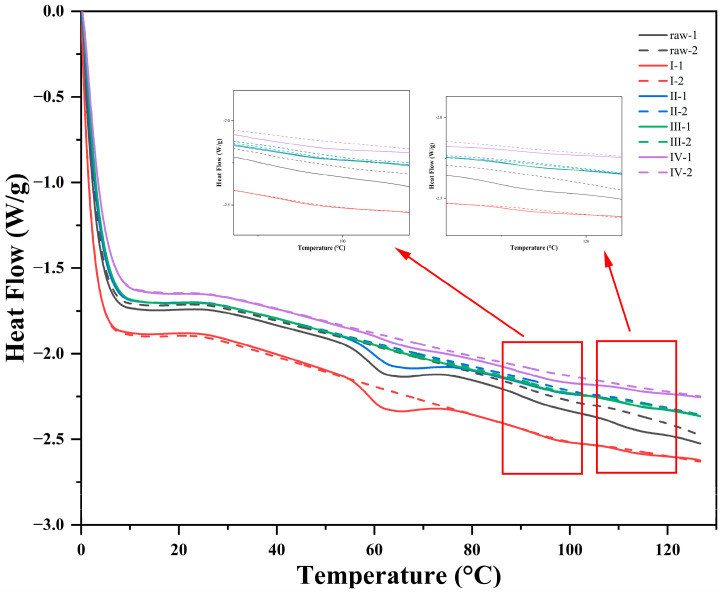
DSC pattern of native oat (raw) in mixing zone (I), cooking zone (II), melting zone (III), and extrudate (Ⅳ) for two scans. Solid lines (1): the first scan; Dotted lines (2): the second scan.

**Figure 5 foods-11-02206-f005:**
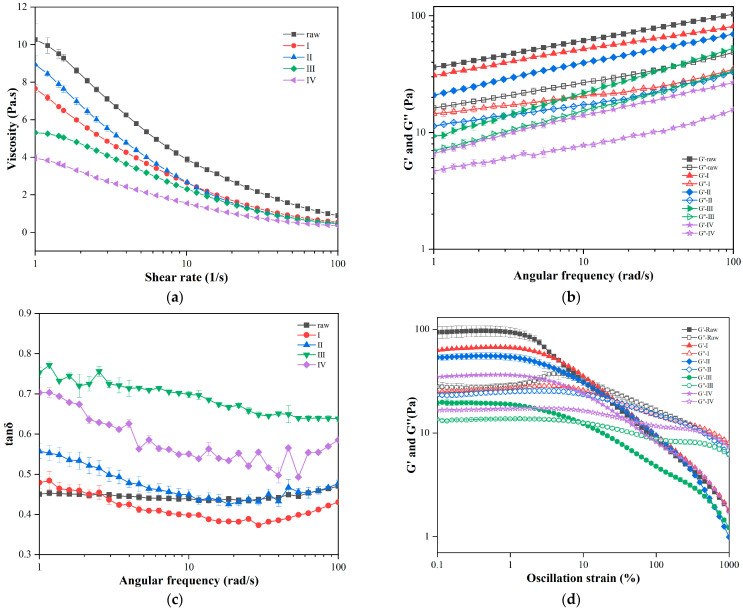
Rheological behavior of oat pasta at 90% moisture: variation of flow behavior with shear rate (**a**), variation of storage modules *G′*, loss modules *G″* (**b**) and *tanδ* (**c**) with frequency, and large amplitude oscillatory shear behavior (**d**) of native oats (raw), in mixing zone (I), cooking zone (II), melting zone (III), and extrudates (Ⅳ).

**Figure 6 foods-11-02206-f006:**
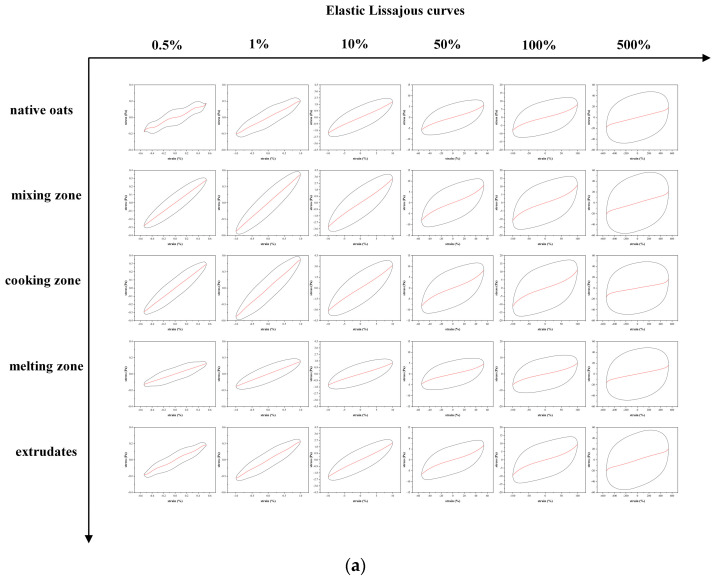
Elastic Lissajous curves (**a**) and viscous Lissajous curves (**b**) of oat paste for native oat (raw), in mixing zone (I), cooking zone (II), melting zone (III), and extrudates (IV) and strain amplitude of 0.5%, 1%, 10%, 50%, 100%, and 500%, respectively.

**Figure 7 foods-11-02206-f007:**
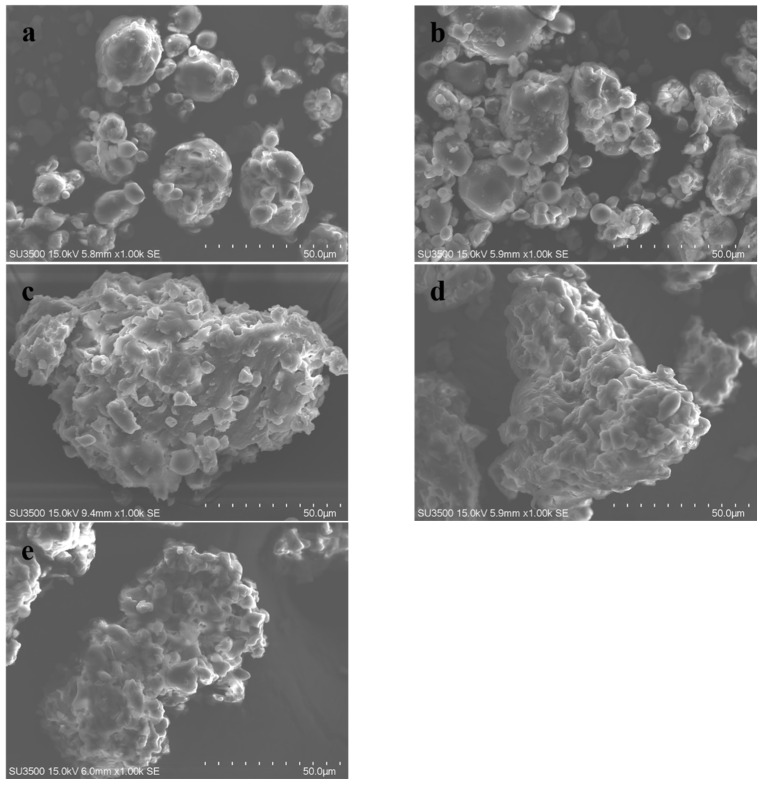
SEM images of native oat (**a**), oat in mixing zone (**b**), cooking zone (**c**), melting zone (**d**), and extrudates (**e**) (×1.00 k magnification).

**Table 1 foods-11-02206-t001:** Proximate composition of native oat flour.

**Ingredients (g/100g)**	**Water**	**Protein**	**Fat**	**Carbohydrate**	**Sodium**
	8.5	12.6	3.1	70.6	0.011

**Table 2 foods-11-02206-t002:** Positions of the diffraction peaks for oats in different extrusion zones and moisture content.

Water Content	Extrusion Zones	Position of the Diffraction Peaks				RC
15%	I	13.82 ± 0.57 ^a^	16.81 ± 0.66 ^a^	18.11 ± 0.35 ^bcde^	21.50 ± 0.68 ^ab^	0.29 ± 0.06 ^a^
	II	13.50 ± 0.19 ^a^	16.63 ± 0.51 ^a^	17.93 ± 0.14 ^de^	21.11 ± 0.19 ^b^	0.33 ± 0.01 ^a^
	III	-	-	17.93 ± 0.05 ^de^	21.00 ± 0.71 ^b^	0.23 ± 0.01^b^
	Ⅳ	-	-	18.15 ± 0.12 ^bcde^	-	0.19 ± 0.02 ^b^
18%	I	13.84 ± 0.90 ^a^	16.78 ± 0.74 ^ab^	19.01 ± 0.37 ^ab^	22.2 ± 0.20 ^a^	0.30 ± 0.02 ^a^
	II	13.97 ± 0.81 ^a^	16.85 ± 0.25 ^ab^	19.25 ± 0.19 ^a^	21.72 ± 0.73 ^ab^	0.30 ± 0.01 ^a^
	III	-	-	18.74 ± 0.79 ^abcd^	21.80 ± 0.28 ^ab^	0.29 ± 0.02 ^a^
	Ⅳ	-	-	19.16 ± 0.12 ^a^	-	0.18 ± 0.01 ^b^
20%	I	14.12 ± 0.22 ^a^	17.04 ± 0.15 ^a^	18.95 ± 0.35 ^abc^	21.97 ± 0.47 ^ab^	0.29 ± 0.03 ^a^
	II	13.75 ± 0.27 ^a^	17.15 ± 0.38 ^a^	19.23 ± 0.14 ^a^	21.93 ± 0.11 ^ab^	0.31 ± 0.01 ^a^
	III	-	-	17.81 ± 0.19 ^e^	-	0.27 ± 0.02 ^ab^
	Ⅳ	-	-	18.03 ± 0.27 ^cde^	-	0.22 ± 0.02 ^b^
NM		13.29 ± 0.13 ^a^	16.12 ± 0.25 ^b^	18.78 ± 0.76 ^abcd^	21.81 ± 0.69 ^ab^	0.28 ± 0.01 ^ab^

NM: native materials; I: oat in mixing zone; II: oat in cooking zone; III: oat in melting zone; IV: extrudate; Different letters above the bars indicate significant differences based on a Duncan test at a level of significance of *p* < 0.05.

**Table 3 foods-11-02206-t003:** Secondary structure of oats protein at 20% moisture.

Extrusion Zones	Secondary Structure (%)			
	α-Helixes	β-Sheets	β-Turns	Random Coils
I	34.49 ± 0.01 ^c^	24.83 ± 0.01 ^c^	26.49 ± 0.02 ^b^	14.19 ± 0.10 ^b^
II	36.10 ± 0.01 ^b^	20.37 ± 0.03 ^d^	28.97 ± 0.07 ^a^	14.56 ± 0.04 ^b^
III	38.18 ± 0.28 ^a^	27.75 ± 0.15 ^b^	15.48 ± 0.02 ^e^	18.54 ± 0.06 ^a^
Ⅳ	31.20 ± 0.59 ^e^	30.02 ± 0.86 ^a^	24.51 ± 0.03 ^d^	14.27 ± 0.29 ^b^
NM	32.79 ± 0.03 ^d^	27.20 ± 0.01 ^b^	25.76 ± 0.07 ^c^	14.25 ± 0.10 ^b^

NM: native materials; I: oat in mixing zone; II: oat in cooking zone; III: oat in melting zone; IV: extrudate; Different letters above the bars indicate significant differences based on a Duncan test at a level of significance of *p* < 0.05.

**Table 4 foods-11-02206-t004:** DSC peak temperatures and enthalpies for oats in four extrusion zones and raw materials.

Samples		Peak 1		Peak 2		Peak 3		Secondary Scanning	
		Peak Temperature (°C)	Enthalpy (J/g)	Peak Temperature (°C)	Enthalpy (J/g)	Peak Temperature (°C)	Enthalpy (J/g)	Peak Temperature (°C)	Enthalpy (J/g)
15%	I	62.24 ± 0.28 ^g^	9.22 ± 0.18 ^a^	97.32 ± 0.12 ^a^	1.84 ± 0.56 ^a,b,c^	115.23 ± 0.05 ^a^	0.51 ± 0.06 ^b,c,d^	100.92 ± 0.07 ^a,b,c^	4.15 ± 0.17 ^d^
	II	63.27 ± 0.31 ^e,f,g^	7.61 ± 0.23 ^b,c^	96.83 ± 0.26 ^a,b^	1.6 ± 0.25 ^a,b,c^	115.63 ± 0.14 ^a^	0.88 ± 0.19 ^a^	101.38 ± 0.38 ^a^	5.54 ± 0.86 ^b,c^
	III	68.61 ± 0.11 ^a,b^	1.06 ± 0.17 ^h,i^	96.56 ± 0.17 ^a,b,c^	0.16 ± 0.03 ^e^	116.31 ± 0.33 ^a^	0.18 ± 0.05 ^f^	98.82 ± 0.05 ^c,d,e^	6.71 ± 0.41 ^a^
	Ⅳ	66.37 ± 3.06 ^b,c,d^	1.57 ± 0.38 ^g,h^	96.12 ± 0.49 ^a,b,c^	0.19 ± 0.1 ^e^	114.84 ± 1.75 ^a^	0.27 ± 0.08 ^d,e,f^	98.85 ± 0.24 ^b,c,d,e^	6.21 ± 0.22 ^a,b^
18%	I	62.49 ± 0.14 ^f,g^	7.03 ± 0.05 ^d^	96.01 ± 0.59 ^a,b,c^	1.38 ± 0.31 ^b,c,d^	116.49 ± 1.64 ^a^	0.63 ± 0.17 ^a,b^	97.99 ± 0.07 ^d,e,f^	4.55 ± 0.25 ^c,d^
	II	64.83 ± 0.95 ^d,e,f^	5.99 ± 0.06 ^e^	97.04 ± 0.26 ^a^	1.03 ± 0.94 ^c,d^	115.31 ± 2.90 ^a^	0.36 ± 0.19 ^b,c,d,e,f^	99.23 ± 2.55 ^a,b,c,d^	1.78 ± 0.08 ^e,f^
	III	67.97 ± 0.10 ^a,b,c^	2.28 ± 0.04 ^f^	95.70 ± 0.35 ^b,c^	0.55 ± 0.3 ^d,e^	115.03 ± 3.37 ^a^	0.35 ± 0.13 ^c,d,e,f^	96.26 ± 0.37 ^f,g^	1.17 ± 0.39 ^f^
	Ⅳ	65.65 ± 0.47 ^c,d^	1.04 ± 0.10 ^i^	96.29 ± 0.95 ^a,b,c^	1.18 ± 0.11 ^c,d^	113.2 ± 0.28 ^a^	0.49 ± 0.01 ^b,c,d,e^	96.06 ± 0.90 ^f,g^	1.68 ± 0.09 ^e,f^
20%	I	62.60 ± 0.49 ^f,g^	8.77 ± 0.04 ^a^	96.78 ± 0.57 ^a,b^	2.31 ± 0.06 ^a^	114.55 ± 0.12 ^a^	0.61 ± 0.07 ^a,b,c^	97.48 ± 0.57 ^d,e,f,g^	3.72 ± 0.22 ^d^
	II	63.35 ± 0.25 ^e,f,g^	7.30 ± 0.28 ^c,d^	96.79 ± 0.50 ^a,b^	1.52 ± 0.19 ^a,b,c^	114.93 ± 0.12 ^a^	0.43 ± 0.16 ^b,c,d,e,f^	101.03 ± 0.28 ^a,b^	1.81 ± 0.08 ^e,f^
	III	69.22 ± 1.46 ^a^	0.72 ± 0.15 ^i^	96.47 ± 0.33 ^a,b,c^	1.25 ± 0.09 ^c,d^	114.15 ± 0.09 ^a^	0.23 ± 0.09 ^e,f^	99.40 ± 0.85 ^a,b,c,d^	2.39 ± 0.67 ^e^
	Ⅳ	66.65 ± 0.63 ^c,d,e^	1.58 ± 0.44 ^g^	96.73 ± 0.17 ^a,b^	2.19 ± 0.35 ^a,b^	115.65 ± 0.03 ^a^	0.31 ± 0.04 ^d,e,f^	96.68 ± 0.24 ^e,f,g^	2.52 ± 0.64 ^e^
NM		62.58 ± 0.35 ^f,g^	7.86 ± 0.03 ^b^	95.37 ± 0.83 ^c^	1.15 ± 0.08 ^c,d^	112.92 ± 1.23 ^a^	0.84 ± 0.01 ^a^	95.74 ± 1.29 ^g^	1.74 ± 0.12 ^e,f^

NM: native materials; I: oat in mixing zone; II: oat in cooking zone; III: oat in melting zone; IV: extrudate; Different letters above the bars indicate significant differences based on a Duncan test at a level of significance of *p* < 0.05.

**Table 5 foods-11-02206-t005:** The model parameters fitting results of flow behavior of native oats and in four extrusion sections.

	Power Law			Cross				
	Viscosity (Pa)	Rate Index	R^2^	Zero-Rate Viscosity (Pa∙s)	Infinite-Rate Viscosity (Pa∙s)	Consistency (s)	Rate Index	R^2^
NM	13.94 ± 0.35 ^a^	−0.58 ± 0.001 ^b^	0.972	15.08 ± 1.68 ^a^	−0.276 ± 0.063 ^e^	0.40 ± 0.079 ^a^	0.68 ± 0.025 ^e^	0.999
I	10.52 ± 1.23 ^c^	−0.63 ± 0.030 ^d^	0.973	8.23 ± 1.30 ^c^	0.140 ± 0.050 ^c^	0.23 ± 0.070 ^c^	0.95 ± 0.070 ^d^	0.999
II	13.26 ± 0.43 ^b^	−0.73 ± 0.003 ^e^	0.978	11.69 ± 0.15 ^b^	0.151 ± 0.001 ^b^	0.37 ± 0.003 ^b^	0.98 ± 0.007 ^c^	0.999
III	8.15 ± 0.04 ^d^	−0.61 ± 0.002 ^c^	0.954	5.45 ± 0.04 ^d^	0.132 ± 0.003 ^d^	0.14 ± 0.001 ^e^	1.04 ± 0.003 ^a^	0.998
Ⅳ	5.16 ± 0.17 ^e^	−0.57 ± 0.001 ^a^	0.966	3.98 ± 0.39 ^e^	0.154 ± 0.012 ^a^	0.17 ± 0.024 ^d^	1.03 ± 0.049 ^b^	0.998

NM: native materials; I: oat in mixing zone; II: oat in cooking zone; III: oat in melting zone; IV: extrudate; Different letters above the bars indicate significant differences based on a Duncan test at a level of significance of *p* < 0.05.

**Table 6 foods-11-02206-t006:** The Power Law model parameters fitting results of frequency dependence of native oats and in four extrusion sections.

	Power Law					
	*K′* (Pa)	*n′*	R^2^	*K″* (Pa)	*n″*	R^2^
NM	36.10 ± 0.89 ^a^	0.23 ± 0.003 ^d^	0.999	15.90 ± 0.85 ^a^	0.23 ± 0.001 ^b^	0.996
I	31.55 ± 0.49 ^b^	0.21 ± 0.001 ^e^	0.997	13.92 ± 0.68 ^b^	0.18 ± 0.012 ^c^	0.977
II	21.45 ± 0.87 ^c^	0.26 ± 0.002 ^c^	0.998	10.91 ± 0.37 ^c^	0.22 ± 0.006 ^b^	0.980
III	9.15 ± 0.61 ^d^	0.38 ± 0.010 ^a^	0.999	6.98 ± 0.32 ^d^	0.34 ± 0.007 ^a^	0.999
Ⅳ	6.90 ± 0.41 ^e^	0.29 ± 0.008 ^b^	0.996	4.55 ± 0.30 ^e^	0.25 ± 0.020 ^b^	0.983

NM: native materials; I: oat in mixing zone; II: oat in cooking zone; III: oat in melting zone; IV: extrudate; Different letters above the bars indicate significant differences based on a Duncan test at a level of significance of *p* < 0.05.

## Data Availability

Not applicable.

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
