# Peer review of "Twin-Screw Extrusion of Oat: Evolutions of Rheological Behavior, Thermal Properties and Structures of Extruded Oat in Different Extrusion Zones"

_foods, 2022, doi:10.3390/foods11152206_

Round 1

Reviewer 1 Report

The manuscript entitled "Twin-screw extrusion of oat: Evolutions of rheological behavior, thermal properties and structures of extruded oat in different extrusion zones" describes the physical and physicochemical changes of dough in the twin screw extruder. The analyses are comprehensive to show the changes during the process and sound.

The differences between extruding phases are discussed for each property, and it is not very clear how properties combinedly show the differences in each stage and overall quality of extruded products. I suggest to rewrite the conclusion section to discuss their findings how each property they observed can contribute the overall changes from flour to the extruded products.

Overall, they should manage to tidy up the writing to clear the arguments to convey their findings to the audience as coordinated arguments between the stages.

Specific comments

1. The significant figures of each measurement

For example, Table 4 gives the peak temperature with the order of 0.01°C. Does it really have such fine resolution and is the difference of less than 0.1°C meaningful to evaluate the property? The instrument may have that fine digit, but it is really matter to show the differences under 0.1°C? Or 0.01% differences in the secondary structure represent something critical? Moreover, protein proximate is 12.6 g while sodium is 11g. It means proteins fell between 12.55 and 12.64 g while sodium was between 10.5 and 11.4. I suggest the authors to pay more attention to the significant figures in their experiments.

2. Colour scheme of the figures

From plot to plot, the authors used different colours to represent one stage. Stage I is mostly represented by red-colour, but Stage II uses blue, cyan, purple, green, and so on. I recommend to use the same shade of colour, i.e., can employ red, wine red, crimson red and so on for the variants within the stage, but all plots across figures should use reddish colour for the particular stage.

Reviewer 2 Report

This work present the Evolutions of rheological behavior, thermal properties and structures of extruded oat in different extrusion zones.  This manuscript need changes which are mention below

Abstract: The abstract should always be concise and informative. The arguments of why your study is important not making any sense. Extensive revision is required in abstract, as the present sentences sounds noisy during reading. There are grammatical errors and need to minimize the sentences in length. Also please delete the first sentence from the abstract. Overall the abstract is not informative enough and need to show the actual picture of the work. Authors are requested to please indicate the numerical values with significant difference in abstract.

Keywords: Need to change the keywords, these keywords are not enough for researching in search engine.

Introduction: Introduction is supported with nice arguments, some information need to address here

1.     Extrusion-cooking need to compare with other alternative processing technologies. Authors are encourage to support their work in introduction section.

2.     I don’t get any information why extrusion cooking have benefits over other methods?

3.     Is extrusion cooking is cost effective? Explain in introduction section

4.     What is the safety and environment concern for extrusion cooking? Indicate in introduction section.

5.     Authors should support their work with arguments why such work is important for food industries?

6.     Please add/indicate and compare your work with pervious published paper.

Methods and materials:

1.     Why the samples were lyophilized? Haven’t control moisture contents during extrusion? Simply can dry in oven?

2.     What was the composition of oat flour after cooking? Haven’t mentioned.

3.     What type of functional groups were intended to see in FTIR? Describe briefly in methods section. Same for DSC and XRD.

4.     Need to define the pasting parameters in details.

5.     What type of curves were intended in rheological parameters determination?

6.     What type model was used to calculate the rheological behavior of oat flour? In dry state or in wet state? Explain

Results and discussion: Discussion upon figures and table are fair and according to findings. Add more of information for XRD and SEM. 

There is no sense to Figure 6 A and B. did not explain anything in first look. How to compare figure 6 A to figure 6B? Clear this situation in discussion part.

Why only 50 µm was selected for SEM images? What about 20? Put at 20 so can detect the actual chemistry

How the peaks were calculated in XRD? Is any reference for peaks calculation?

Conclusion: Change the conclusion as per changes in discussion, add more of application for this research for common reader.

Round 2

Reviewer 2 Report

I still suggest to please minimize the abstract. Also need correction in sentences. 
